# TCL1A, B Cell Regulation and Tolerance in Renal Transplantation

**DOI:** 10.3390/cells10061367

**Published:** 2021-06-01

**Authors:** François Brinas, Richard Danger, Sophie Brouard

**Affiliations:** 1Université de Nantes, CHU Nantes, Inserm, Centre de Recherche en Transplantation et Immunologie, UMR 1064, ITUN, F-44000 Nantes, France; Francois.Brinas@etu.univ-nantes.fr; 2Labex IGO, F-44000 Nantes, France; 3Centre d’Investigation Clinique en Biothérapie, Centre de Ressources Biologiques (CRB), Labex IGO, F-44000 Nantes, France

**Keywords:** TCL1A, kidney transplantation, tolerance, Breg, Akt, IL-10

## Abstract

Despite much progress in the management of kidney transplantation, the need for life-long immunosuppressive therapies remains a major issue representing many risks for patients. Operational tolerance, defined as allograft acceptance without immunosuppression, has logically been subject to many investigations with the aim of a better understanding of post-transplantation mechanisms and potentially how it would be induced in patients. Among proposed biomarkers, *T-cell Leukemia/Lymphoma protein 1A* (*TCL1A*) has been observed as overexpressed in the peripheral blood of operational tolerant patients in several studies. TCL1A expression is restricted to early B cells, also increased in the blood of tolerant patients, and showing regulatory properties, notably through IL-10 secretion for some subsets. *TCL1A* has first been identified as an oncogene, overexpression of which is associated to the development of T and B cell cancer. TCL1A acts as a coactivator of the serine threonine kinase Akt and through other interactions favoring cell survival, growth, and proliferation. It has also been identified as interacting with others major actors involved in B cells differentiation and regulation, including IL-10 production. Herein, we reviewed known interactions and functions of TCL1A in B cells which could involve its potential role in the set up and maintenance of renal allograft tolerance.

## 1. Introduction

Recently, transplantation has become the best treatment for end-stage renal disease. Continuous improvement in transplantation outcomes is due in part to better clinical management in the instauration of life-long immunosuppressive (IS) therapy, which is required to control recipients’ alloimmune response and prevent graft rejection [1]. In addition, many attempts have been made to minimize IS treatments [2], which are still sources of important side effects, including infection, cancer, nephrotoxicity and metabolic complications [3,4,5,6] with insufficient benefits for long-term graft survival [7]. In this context, tolerance defined as allograft acceptance in the absence of immunosuppression represents the ultimate goal in solid organ transplantation, as it avoids IS treatment side effects and thereby improves the recipient’s quality of life. Many attempts have been made in the field of tolerance induction since the first report of Billingham et al. [8,9,10,11,12,13,14]. In addition, the literature has also described patients who have shown graft stability even after a prolonged period without any IS or protocol for tolerance induction, although their proportion among transplantation patients is small [15,16,17,18]. These patients are considered as “operationally tolerant”, a state defined on the basis of the clinical criterion of stable graft function in the absence of any immunosuppressive treatment for more than 1 year [17,18]. Immunosuppression withdrawal can result from spontaneous interruption in treatment, such as that caused by noncompliance of patients or medical decisions, especially in the case of lymphoproliferative posttransplantation disorders [17]. In kidney transplantation, no clinical parameter has been found as associated with this state, but operationally tolerant patients remain immunocompetent and do not show greater susceptibility to opportunistic infections than healthy volunteers [17,19]. Few cases have been reported in kidney transplantation with an incidence that has been estimated inferior to 5% [16,17,20]. On the other hand, in liver transplantation, an operationally tolerant state is more likely, as shown by immunosuppression-weaning protocols’ success rates of more than 40% with selected patients [21,22].

Hence, operationally tolerant patients have been subject to many investigations in recent decades with the aim of better understanding tolerance mechanisms and/or predicting when these mechanisms can be triggered. In the absence of graft biopsy in these operationally tolerant patients (TOL), investigations have focused on transcriptional and phenotypic analyses in peripheral blood. One objective is to design a tolerance-specific signature suitable for identifying patients likely to benefit from safe immunosuppression minimization or for identifying potent pharmacological targets for tolerance induction therapies [23,24,25,26,27,28,29,30,31,32,33].

## 2. TCL1A, a B Cell Biomarker of Tolerance Process

Even though little overlap of biomarkers has been found between all these studies, as illustrated by the meta-analysis of Baron et al. [31], some recurrent transcripts have been identified as tolerance markers, including *TCL1A*. The *TCL1A* gene was first identified as a potential biomarker of tolerance in kidney transplantation, being overexpressed in peripheral blood mononuclear cells (PBMC) of TOL compared to healthy volunteers (HVs), by 1.80 fold [23]. In 2010, Newell et al. reported that *TCL1A* was overexpressed with a mean change greater than 2-fold between TOL and stable patients with immunosuppression (STA) [25], and Sagoo et al. also found that two different probes targeting *TCL1A* were significantly overexpressed in TOL patients compared to STA and patients under chronic rejection (CR) (median log-fold change relative to the median of all samples with probe 1: TOL = 1.062, STA = −0.41, CR = 0.0965 and with probe 2: TOL = 0.994, STA = −0.38, CR = 0.121). This group also reported that both probes targeting *TCL1A* were part of the top 5 significant among 170 differentially expressed genes in two independent cohort of patients. These results were confirmed by qRT-PCR [27]. *MS4A1* gene coding for CD20 was also found as differentially over-expressed in these TOL, evidencing a B cell-related signature including *TCL1A* (23–25, 27). To find a common signature in TOL from these studies with a relatively low number of samples, Baron et al. performed a meta-analysis of previous studies by normalizing and merging their transcriptome data. Logically, *TCL1A* was found in the top 20 gene signatures that may discriminate TOL from STA patients with 91.7% accuracy, along with some B-cell related genes such as *MS4A1, CD40* and *C79B* [31]. Finally, this signature was refined in a composite signature of six genes, including *TCL1A* (plus *AKR1C3, CD40, CTLA4, ID3* and *MZB1*) and two clinical parameters that enabled the discrimination of TOL (42 TOL; 189 STA) with an area under the curve (AUC) of 0.973 and with negative and positive predictive values of 0.989 and 0.800, respectively [33].

These studies highlight the fact that TCL1A is a potent marker of tolerance in kidney transplantation but do not postulate on its potential role. As we shown later in the paper, *TCL1A* expression is closely related to B cells, specifically to early B cells before they move through germinal centers (GCs) [34,35]. Common augmentation patterns of total B cells and their proportion have been found in the peripheral blood of tolerant patients [25,26,27,36] in parallel with a transcriptional signature of tolerance that comprises B cell-related markers [23,24,25,33]. Studies reported phenotypes of early B cell populations and increased levels of naïve B cells (CD19^+^; CD20^+^; CD27^−^; IgM^+^; IgD^+^; CD24^low^; CD38^low^) and transitional B cells (CD19^+^; CD20^+^; CD27^−^; IgM^+^; IgD^+^; CD24^high^; CD38^high^) [25,26,30,36,37] associated with a decrease in plasma cells (CD19^+^; CD38^+^; CD138^+^) in the peripheral blood of TOL patients [30]. These studies indicate a higher proportion of immature than differentiated and activated B cells in the blood of TOL patients, which is consistent with the selective expression of TCL1A.

While B cells were initially considered in the context of renal transplantation as antibody-secreting cells, it has also been established that they have effects as antigen-presenting cells and cytokine producers [38,39]. Some B cell subsets with phenotypic diversity resulting from multiple stimuli (TLR [Toll-like receptor]; CD40; B-cell receptor [BCR]) have been identified [40,41], implying downregulation of the effector immune response in different contexts and through several mechanisms, including anti-inflammatory cytokine production, such as interleukine-10 (IL-10) [40,42] and transcriptional growth factor β (TGF-β) [41,43]. When studying the B cell phenotype of kidney transplant patients, Pallier et al. found that TOL patients presented an increased frequency and absolute value of B cells expressing CD1d^+^ and CD5^+^ compared to STA and CR patients [26]. Bhan and collaborators were the first to report the involvement of a B cell subset expressing high levels of CD1d and able to regulate Th2-mediated inflammation in a colitis mouse model through IL-10 production [44]. Another study highlighted the ability of a mouse model with a small spleen population of CD1d^hi^ CD5^+^ B cells to reduce the contact hypersensitivity response through IL-10 production. These splenic IL-10-producing CD1d^hi^ CD5^+^ regulatory B cells (Breg) cells were described as murine ‘‘B10 cells’’ [45]. The CD1d^+^ CD5^+^ B cells found by Pallier et al. in blood from TOL patients exhibited a phenotype similar to that of the mouse B10 cells. However, the IL-10 expression level in these human B cells was not different between different groups of patients after stimulation [26]. Blair et al. characterized additional human Breg subsets. They found that immature CD19^+^ CD24^hi^ CD38^hi^ transitional B cells in the peripheral blood of HVs showed the greatest ability to produce IL-10 following stimulation and were able to suppress T cell proliferation in coculture assays in an IL-10 level-dependent manner [42], which was confirmed in further studies [46,47]. This immature population corresponded to populations described in several studies as increased in blood obtained from kidney transplanted TOL patients [25,30,36,37]. Additionally, Blair et al. found that a large proportion (71%) of the CD1d^hi^ CD5^+^ B cell subset identified by Pallier et al. was also evident in the CD19^+^ CD24^hi^ CD38^hi^ B cell population, indicating that it would correspond to a same Breg subset [26,42]. To more deeply characterize the function of these B cells, Newell et al. measured IL-10 expression in B cells following stimulation and found a significant, although low, increase in IL-10 expression in T1 and T2 transitional B cells (CD38^+^CD24^+^) in TOL samples compared to its expression in HVs and STA kidney transplanted patients [25]. Similarly, Chesneau et al. found a significant increase in IL-10 production after a 4-day culture of B cells from TOL patients, compared to those from HVs and STA kidney transplanted patients, which was confirmed by intracellular staining [30]. These results matched Blair et al.’s observations [42] and were further confirmed by Cherukuri et al., who found that transitional CD24^hi^ CD38^hi^ B cells were the most IL-10 competent B cell subset and that these cells showed a decreased IL-10/TNFα ratio in patients with kidney graft dysfunction [48,49]. Nova-Lamperti et al. also recently showed that in kidney transplant patients, BCR signaling was reduced in TOL patients, compared to HVs, with a reduction in ERK phosphorylation, a downstream effector of mitogen-activated kinase (MAPK) that participates in the reduction of IgG1 levels in TOL patient serum, indicating a lack of B cell differentiation [30,50]. The same authors also pointed out that CD40 stimulation induced IL-10 production in transitional B cells in HVs and TOL patients, whereas dual stimulation of BCR+CD40 cells led to downregulated IL-10 production in HVs but not in TOL patients, indicating that reduced BCR signaling leads to IL-10 production by transitional B cells preferentially in TOL patients [50]. Taken together, these studies show that TOL have more peripheral B cells, specifically, more immature B cells, whose phenotype has been characterized as immunoregulatory, with a concordant transcriptional signature composed of several B cell-related genes, including *TCL1A*.

As indicated by Rebollo-Mesa et al., immunosuppressive therapies (especially prednisone and azathioprine) strongly impact the expression in kidney transplant patients of B cell-related genes and B cell subpopulations, particularly transitional CD24^hi^ CD38^hi^ B cells [51]. The increase in transitional B cells in TOL patients may be the result of immunosuppression weaning, although it does not necessary preclude a role for this B cell subpopulation in this process [51]. Christakoudi et al. recently attempted to identify a tolerance signature by taking into account the impact of immunosuppression using statistical adjustments [52]. They report that, even after statistical adjustments considering IS treatments, the expression level of *TCL1A* can be used to discriminate TOL from non-TOL patients [52]. Granata et al. also reported a decrease expression of *TCL1A* in total PBMC, isolated T cells and monocytes after 6 h in vitro incubation with everolimus, while it was overexpressed in B and T cells with high dose of tacrolimus, compared to cells from healthy individuals [53]. Finally, *TCL1A* is not merely a marker of renal transplantation tolerance. Viklicky et al. reported that *TCL1A*, *MS4A1* and *CD79B* were overexpressed in the blood of STAs kidney recipients during the first 6-month follow-up period compared to patients with a rejection episode [29]. Similarly, Heidt et al. reported increased *TCL1A* gene expression in blood and isolated B cells during the post-transplantation period in STA kidney transplant patients, but that it was profoundly decreased in patients with acute rejection. A receiver operating characteristic (ROC) curve analysis showed a high discriminative capacity for predicting acute rejection by *TCL1A* (AUC = 0.86, *p* < 0.001), and higher than for *CD79B* (AUC = 0.76, *p* = 0.01) [32]. In lung transplantation, Danger et al. also described that the decrease of *TCL1A* blood expression predicts the chronic lung allograft dysfunction, the main cause of lung allograft loss [54]. *TCL1A* was, again, clustered with B cell-related genes such as *CD19, MS4A1, BANK1* and *CD40*.

TCL1A thus appears to be a strong marker for both immune system quiescence and rejection episodes after allograft transplantation. These studies evidenced *TCL1A* within B cell-related signatures, in addition to genes such as *MS4A1, CD40* and *CD79B*. However, how TCL1A is involved in B cell regulation and tolerance maintenance remain to be deciphered.

## 3. TCL1A and B Cell Biology

Human TCL1A is a 14 kDa protein containing 114 amino acids and shares 50% identity with murine TCL1A [55,56,57]. TCL1A encodes a small intracellular, nonenzymatic protein that acts as a coactivator of protein kinase B/Akt, a serine-threonine kinase central to many signaling pathways, with multiple effects, including cellular proliferation, growth and survival [58,59]. TCL1A structure is composed of a hydrophobic core surrounded by two 4-stranded β-sheets. One sheet is composed of shorter strands (βA-βB and βE-βF) and the other of longer strands (βC-βD and βG-βH). The two sheets are linked by a long and flexible loop between strands βD and βE, with an overall organization of a β-barrel [56]. TCL1A contains a homodimerization domain in the βc strands, which is required for effective Akt coactivation. Once dimerized, TCL1A can bind the three Akt isoforms through an interaction between the βa, βb, βe and βf strands with two Akt PH domains [56,60,61]. In B cells, the Akt signaling pathway is triggered by the activation of tyrosine kinase receptors, in particular BCR, inducing a phosphorylation cascade involving multiple agents and allowing Akt recruitment to the plasma membrane as mediated by phosphatidylinositol 3-kinase (PI3K) and full activation by 3-phosphoinositide-dependent protein kinase 1 (PDK1), rapamycin-insensitive mTOR complex 2 (mTORC2) and other kinases [62,63,64]. Once activated, Akt detaches from the membrane and activates downstream pathways. In the TCL1A-Akt dimer configuration, TCL1A does not directly phosphorylate Akt but facilitates Akt interactions and phosphorylation at the membrane, blocking its inactivation and augmenting Akt activation, and promoting its translocation to the nucleus (cf. Figure 1) [58,60,61,63].

As reviewed recently by Paduano et al., TCL1A has been found to interact and regulate many pathways independent of its interaction with Akt [65]. Notable interaction partners include DNA methyltransferase A (DNMT3A) and B [66], receptor tyrosine kinase-like orphan receptor 1 (ROR1) [67], and heat shock protein 70 (Hsp70) [68]. These Akt-independent interactions have been characterized as impacting TCL1A-induced oncogenesis but may also be involved in other processes.

TCL1A was first identified as an oncogene involved in the development of T-cell leukemia [35,69,70]. In humans, TCL1A is located on chromosome 14q32.1 and belongs to a family consisting of five genes, *TCL1A*, *TCL1B*, *TNG1*, *TNG2* and *MTCP1* [71,72,73]. TCL1A is expressed in the fetal liver, fetal spleen, fetal kidney, tonsils, testis and peripheral blood lymphocytes [72,73,74]. Its expression has also been detected in plasmacytoid dendritic cells [75]. Under physiological conditions, TCL1A is implicated in embryogenesis and lymphopoiesis [74]. Only immature CD4^+^ CD8^+^ T cells express TCL1A, which is silenced in mature T cells [35]. In B cells, TCL1A expression is primed in pro-B cells with a peak intensity in pre-B cells and persists in IgM^+^ naïve B cells. It is also expressed in mantle zone B cells but strongly repressed in marginal zone and germinal center (GC) B cells and completely silenced in post-GC memory and plasma cells [34,35,76]. Low expression levels have been detected in differentiated plasmacytoid cells in vitro [34]. Dysregulated TCL1A expression in animal models impairs the ability of an embryo to develop toward the morula stage [74]. It also leads to impaired lymphopoiesis and increased sensitivity to receptor-mediated apoptosis in immature cells undergoing positive and negative clonal selection, underlying the major role of TCL1A in lymphocyte development [76]. In contrast, TCL1A overexpression is identified as a cause of T and B lymphocyte malignancies in humans. The TCL1A gene has been discovered in thymocytes in abnormal chromosomal rearrangements with inversion or translocation of TCL1A with either the α/δ or β-chain locus of the T cell receptor (TCR). This leads to its overexpression through TCR enhancers and the development of T cell leukemias [70,77]. TCL1A overexpression also occurs in B lineage cells and can lead to multiple types of lymphoma arising from pre-GC differentiation stages [78,79].

The mechanisms of TCL1A-mediated lymphomagenesis and its post-GC repression are not yet fully understood. The smallness of the group of cells expressing TCL1A seems to indicate that physiological regulation and pathological overexpression in B cells may occur at the transcriptional level [35,76]. Several cis-regulatory elements have been identified in the *TCL1A* promoter. The transcription factor Nur77 may regulate *TCL1A* expression through the nerve growth factor-responsive element in a negative retroactive loop that is dependent on Nur77 phosphorylation by Akt [80]. Researchers have identified Sp1-binding sites in the *TCL1A* core promoter region and demonstrated its implication in *TCL1A* transactivity in vivo [81]. Other mechanisms for TCL1A regulation have been proposed, such as regulation by microRNAs miR-29 and miR-181 [82,83] or epigenetic modification [84,85], but even taken together, these proposed mechanisms cannot fully explain physiological *TCL1A* repression or pathological overexpression.

Kuraishy et al. identified a CREB responsive element (CRE)-like half site within the *TCL1A* promoter region activated by phosphorylation of its Ser-133 (pCREB-133). CREB activation is mediated by cAMP and PKA activation [86]. In HEK293 cells, TCL1A expression is cAMP-dependent. Inducing CREB overexpression with a transgene induced a 3-fold increase in TCL1A expression, which was not observed in cells with a CRE half-site mutation [85]. TCL1A expression was not dependent on pCREB-133 but was dependent on cooperation between CREB and transducer of regulated CREB protein 2 (TORC2) or CRTC2 [87]. AICAR (an AMP analog) induced CRTC2 phosphorylation and inactivation, leading to dose-dependent repression of *TCL1A*, indicating collaboration between CREB and CRTC2 to induce TCL1A expression. Moreover, CRTC2 siRNA induced 40% *TCL1A* repression in Nalm6 cells. These results show that CRTC2 could be involved in the physiological and pathological regulation of TCL1A in B cells [85].

Interestingly, both BCR and CD40 stimulation induces CRTC2 cytoplasmic retention and TCL1A repression and increases cell apoptosis, whereas cAMP allowed translocation of CRTC2 to the nucleus, restored TCL1A expression and prevented the apoptosis of Ramos cells [85]. These results suggest that TCL1A expression relies heavily on BCR and CD40 stimulation, with its repression beginning in GC and relying to a certain extent on CRTC2 phosphorylation, with a signaling hierarchy that notably implicates cAMP.

## 4. TCL1A and B Cell Survival

The PI3K-Akt signaling pathway axis exhibits many functions in B cells, notably promoting survival through several target and transcription factors [88,89,90,91,92,93,94]. Multiple pro-survival signaling pathways and TCL1A-Akt coactivation may partially explain the impact of B cell overexpression of TCL1A on their oncogenesis [35,84,95]. In addition, Akt has been characterized as preferentially inducing the expression of Bcl-2 antiapoptotic family member Mcl-1 upon IL-3 stimulation, with transcription depending in part on cAMP response element-binding protein (CREB) binding to CRE-2 in the Mcl-1 promoter region [96,97]. In 2009, Tabrizi et al. focused on the role of the BCR signaling cascade in the apoptosis of B cells with naïve or memory phenotypes. After 2 days of culture, both subsets underwent spontaneous apoptosis, with BCR stimulation rescuing the naïve B cells but not the memory B cells. Following BCR stimulation, high Mcl-1 expression and a 10-fold increase in TCL1A mRNA level were found in the naïve B cells but were unchanged in the memory B cells. After inducing its expression in memory B cells through transgene activation, TCL1A restored Mcl-1 expression and protected memory B cells from apoptosis. These results indicate that TCL1A is implicated in Mcl-1 expression and differentially regulated in early and mature B cells at the transcriptional level, depending on BCR stimulation. Nonetheless, TCL1A protein was only detected in the naïve B cells, and irrespective of BCR stimulation its expression decreased over time, suggesting potent posttranscriptional regulation of TCL1A expression [98]. Opposite effects have been reported for Galectin-1 protein (Gal-1) and TCL1A expression, with distinct BCR signaling that was reduced in naïve B cells compared to memory B cells, leading to TCL1A expression and the survival of one B cell subset and to Gal-1 expression and apoptosis for another B cell subset [98]. This result was similar to that of reduced BCR signaling in TOL patients with reduced ERK phosphorylation, which led to preferential IL-10 production upon BCR+CD40 stimulation [50]. This TCL1A effect on early B cell survival was also consistent with their increased proportion in TOL patients.

Perkarsky et al. were the first to characterize NF-κB enhancement by TCL1A independent of Akt [99]. Indeed, TCL1A interacted with IκBα, participating in the IκB-NF-κB complex and allowing NF-κB sequestration and inactivation in the cytoplasm [100,101]. TCL1A also physically interacts with ATM (ataxia telangiectasia mutated), a serine-threonine kinase triggered by DNA strand breaks [102]. In Eµ-TCL1A mice, TCL1A overexpression led to upregulated ATM, a protein known as a tumor suppressor, and downregulated TCL1A in the DNA damage repair process in B cells undergoing GC transformation [103,104]. In TCL1A-Transgenic (Tg) mice, ATM upregulation was also associated with IκB downregulation, which may have been the result of IκB phosphorylation by ATM [104]. ATM also inhibits apoptosis in acute myeloid leukemia and myelodysplastic syndrome by increasing IκB degradation, favoring NF-κB activation [105]. The authors hypothesized that TCL1A interaction with both IκBα and ATM may have increased ATM-mediated degradation of IκBα, which may partially explain TCL1A Akt-independent NF-κB activation, reinforcing its impact on B cell survival [104]. Moreover, Gaudio et al. recently showed that TCL1A interacted with the AP-1 consensus region of the *TP63* promoter, an oncogene implied in lymphomagenesis [106]. In vitro, TCL1A expression enhanced the transcriptional activity of the *TP63* promoter, and inhibition of the expression of both proteins markedly reduced Akt phosphorylation and Raji cell survival. These results revealed another interaction by which TCL1A may prevent B cell apoptosis [107].

## 5. TCL1A, Cell Proliferation and Differentiation

Cell growth and proliferation are positively regulated by the PI3K-Akt pathway. Akt might promote proliferation by inhibiting GSK3 (glycogen synthase kinase 3) activity, preventing ubiquitination and degradation of CyclinD and c-Myc [108,109]. Akt also directly activates mTORC1, whose serine/threonine kinase activity promotes growth and biogenesis through activation of the p70 S6 kinase and inhibition of the translational repressor 4E-BP1 [110,111]. Different models have been used to decipher the role of TCL1A in these processes. Eµ-TCL1A transgenic mice in which the *TCL1A* gene was under the control of a VH promoter-IgH-Eµ enhancer exhibited TCL1A expression enhancement in immature and mature B cells, serving as a model of human B chronic lymphocytic leukemia (B-CLL) physiopathology. These mice showed polyclonal expansion of CD5^+^ IgM^+^ B220^int^ B cells in primary and secondary lymphoid tissues at approximately 4 and 8 months of age [112,113,114]. In a mouse model of GC-derived lymphoma, there was an increase in the proliferation of B cells under BCR stimulation compared to that in wild-type mice [115]. In another TCL1A-Tg mouse model (Eµ-TCL1FL), B cells developed a CD5^+^ B220^+^ cancerous phenotype at approximately 16-20 months of age. In this model, TCL1A-Tg B cells displayed increased proliferation compared with wild-type B cells, even without stimulation. The level of phospho-Akt was significantly higher in the malignant and total B cells in TCL1A-Tg mice than in the B cells of the spleen in wild-type mice with or without PMA stimulation [83]. Similarly, in cultures of primary B-CLL cells under several stimulation conditions, BCR responsiveness was positively correlated with TCL1A expression, with a robust growth response associated with fast and strong Akt phosphoactivation in B-CLL cells with high TCL1A levels. These cells also presented robust growth in vivo and poor response to therapies [116]. These findings are consistent with observations of TOL patients in which high TCL1A expression is associated to an increase in total B cell count and high early B cell proportion.

TCL1A expression may influence B cell differentiation through several factors. The expression of activation-induced deaminase (AID) is critical for class switch recombination (CSR) and somatic hypermutation (SHM) of immunoglobulins (Ig) in B cells [117]. *IRF4* and *PRDM1*, which encode BLIMP-1, are two major genes involved in B cell maturation, notably in Ig production, and are mandatory for B cell differentiation into plasmablasts and plasma cells [118,119]. IRF4 and BLIMP-1 also both upregulate *XBP1*, another mandatory gene for Ig production is implicated in the endoplasmic reticulum stress response and the establishment of B cell Ig secretion [118,119,120]. XBP-1 was also implicated in the effectiveness of BCR signaling and may upregulate AID expression [121]. Strong PI3K-Akt signaling in B cells favored their early differentiation into antibody-secreting cells, not into class switch antibody secreting cells [122,123]. This fate was notably influenced by Akt activation, which increased BLIMP-1 expression, acting itself as an AID repressor [122,123], and through other factors [122,124,125,126]. Using an Eµ-TCL1A transgenic mouse model and 3 days of LPS stimulation, Kriss et al. showed that CD5^-^ B220^+^ cells from 2-month-old mice overexpressing TCL1A presented increased expression of XBP-1, PAX5 and AID [114], in accordance with increased expression of Aicda gene, coding for the AID protein previously identified in the Eµ-TCL1A mice [115]; these mice also presented reduced expression of IRF4 and BLIMP-1 compared to wild-type mice at 2 months. After 8 months, the transgenic mice developed a CD5^+^ B220^+^ B cell cancer phenotype, and ectopic PAX5 expression and a profound increase in IRF4 and BLIM-1 expression were evident compared to the corresponding levels in the wild-type mice. However, despite this expression profile, these mice did not acquire a CD138^+^ B cell immunophenotype, indicating a defect in late B cell differentiation. These cells also had increased XBP-1 and AID expression compared with CD5^-^ B220^+^ B cells. Surprisingly, after 8 months, these CD5^+^ B220^+^ B cells also showed a substantial reduction in Akt expression, with altered BCR signaling, which was becoming constitutively active, and the BCR transducer CD79A being phosphorylated without the BCR stimulation associated with reduced p-Akt and p-ERK expression [114]. This dysregulated and persistent BCR signaling was consistent with findings on TCL1A-mediated oncogenesis [116,127]. This outcome may have been linked to XBP-1 overexpression since a lack of XBP-1 has been associated with default BCR activation with reduced CD79a, CD79b and Syk phosphorylation [121]. Furthermore, in addition to increasing XBP-1 expression, TCL1A could interact directly with XBP-1, leading to questions on its role in CLL development [114]. In that respect, TCL1A-mediated B-CLL development begins by an increased XBP-1 and AID expression, evolving in 8 months to malignant cells with increased IRF4 and BLIMP-1 expression, associated with reduced Akt expression, which may indicate other ways for TCL1A than Akt coactivation in CLL development (cf. Figure 1) [114].

In GC, AID promotes CSR by generating DNA strand breaks in cells undergoing high rates of proliferation [117]. Sheirman et al. showed that DNA double-strand breaks reduced the association of the TCL1A transcription factor CRTC2 with the *TCL1A* promoter and repressed TCL1A expression [85,103]. This process relied on ATM activation of LKB1, a kinase mediating the activation of AMP-activated protein kinases, including SIK kinase, which is known to phosphorylate and inactivate CRTC2 (cf. Figure 1) [103]. They showed that CRTC2 upregulates several genes in GC B cells that are downregulated in post-GC B cells, including *MYC* and *AID* [103,120], two genes that were overexpressed in TCL1A-Tg mice [114,115]. CRTC2 inactivation was required for plasma cell differentiation since CRTC2 overactivation decreased BLIMP-1 expression and promoted the expression of B cell proliferation factors such as BCL6 [103]. A similar effect on BLIMP-1 expression was found in 2-month CD5^-^ B220^+^ TCL1A-Tg mice [114]. These results confirm the implication of TCL1A and other co-expressed genes in B cell proliferation and point to the need for their repression to promote B cell differentiation starting during GC transformation.

## 6. TCL1A and IL-10, Factors of B Cell Regulation

In naïve B cells, IL-10 production is first initiated by TLR stimulation, allowing further robust IL-10 production through both BCR and CD40 stimulation [128,129,130]. IL-10 expression through TLR4 stimulation requires Syk, an essential kinase in the BCR signalosome. Syk deficiency causes dysregulated TLR4-dependent IL-10 secretion and a lack of CD1d^+^ CD5^+^ IL-10 competent B cells. IL-10 production following TLR4 stimulation requires the presence of both BCR and Syk, leading notably to Akt activation [131]. Adding the PI3K inhibitor LY290042 or the Akt inhibitor triciribine to culture medium significantly reduced IL-10 production in B cells in vitro. Mice deficient in phosphatase and tensin homolog (PTEN), a known inhibitor of the PI3K-Akt pathway, harbored a markedly increased proportion of B10 cells compared to wild-type mice and exhibited significantly reduced contact hypersensitivity and an increased proportion of regulatory T cells in lymphoid organs [132]. Taken together, these findings indicate that the PI3K-Akt pathway may participate in IL-10 B cell competency and reinforce the postulate suggesting that increased TCL1A expression favors B10 cell development. However, the mechanisms by which the PI3K-Akt pathway may promote IL-10 expression are unclear, since the activation of this pathway could interfere with the function of many factors known to govern IL-10 expression in B cells, including STAT3, IRF4, NFAT or C-Maf [130,133,134,135]. GSK3, inhibited directly by the PI3K-Akt pathway [136], has been suggested to play an important role in tolerant phenotype development in several hematopoietic cell subsets. In several T cell subsets, GSK3 inhibition led to promoted IL-10 expression through methylation changes in the *IL10* promoter region and increases in IL-10 transcription factors such as Sp1, c-Maf and BLIMP-1 [137,138]. Moreover, it has been shown that in B cells, isoprenylation factors, notably geranylgeranyl pyrophosphate, and the enzymatic activity of geranylgeranyltransferase (GGTase) were required for effective TLR9-dependent IL-10 production in several B cell subsets [139]. This TLR9 pathway is Ras-dependent and implies the PI3K-Akt pathway. GGTase inhibition reduced IL-10 expression and increased GSK3 activation, while direct GSK3 inhibition restored IL-10 expression in B cells. Similar to its effect in T cells, GSK3 inhibition allows the expression of BLIMP-1, which was identified in a previous study as an important transcription factor for IL-10 in activated B10 cells (cf. Figure 1) [140]. BLIMP-1 was highly upregulated by TLR9 stimulation in several B cell subsets, including naïve and CD24^hi^ CD38^hi^ B cells, with siRNA-mediated BLIMP-1 knockdown reducing B cell IL-10 expression by 50 to 90% [139]. These findings are very important as they explain PI3K-Akt-mediated IL-10 production in B cells and reveal a potential link between TCL1A and IL-10 B cell competency. In addition, GSK3 is implicated in the expression and signaling of IL-10 in multiple hematopoietic cell lineages, representing an important checkpoint for cell phenotype orientation [141,142]. Interestingly, the CREB/CRTC2 complex implicated in B cell TCL1A expression also participated in IL-10 expression in dendritic cells in response to zymosan and autocrine production of prostaglandin E2 [143]. CRTC3 has also been identified as an IL-10 transcription factor in macrophages [144,145]. As previously stated, CRTC2 and CRTC3 are inactivated by SIK kinases, which are inactivated by PKA and activated by GSK3 (cf. Figure 1) [146,147,148,149].

Given the extent to which TCL1A has been implicated in several human B cell CLLs [78,116], Dilillo et al. tested whether TCL1A expression is correlated with IL-10 expression and, using samples from 52 patients with overt CLL, found that TCL1A expression positively correlated with IL-10 expression. Moreover, TCL1A expression was significantly higher in the IL-10^+^ than in the IL-10^-^ CLL cells from the same patients, indicating that TCL1A overexpression may play a role in the development of IL-10 competency in CLL cells [113]. To investigate this question in depth, these authors studied IL-10 competency in CLL cells obtained from Eµ-TCL1A mice. The vast majority (50–90%) of the B cells from Eµ-TCL1A mice older than 12 months were IL-10 competent, while the proportion of IL-10 competent B cells in the wild-type mice was not increased with age. However, some TCL1A-Tg mice through to 16 months of age retained normal frequencies and absolute numbers of IL-10^+^ B cells, demonstrating that TCL1A expression alone is not sufficient to induce B cell IL-10 competency [113]. In vivo, in aged TCL1A-Tg mice, low-dose LPS treatment induced high levels of IL-10 production, with an IL-10 level 159-fold higher in the TCL1A-Tg mice than in the wild-type mice. In vitro, TCL1A-Tg CLL cells possessed regulatory properties and could significantly inhibit macrophage TNF-α production upon activation in coculture assays [113]. These results indicate that sustained TCL1A expression may favor the orientation of B cells to acquire a regulatory phenotype through IL-10 expression.

Studying Eµ-TCL1A mice, Kriss et al. showed that, on the one hand, CD5^-^ B220^+^ cells from 2-month-old TCL1A-Tg mice that had not yet developed a cancerous phenotype presented reduced expression of IRF4 and BLIMP-1 compared to those from wild-type mice after 3 days of LPS stimulation. On the other hand, with the same stimulation, the CD5^+^ B220^+^ cells of 8-month-old mice, corresponding to the development of IL-10 cell competency [113], presented markedly increased expression of these two transcription factors compared to those of wild-type mice, which seems to confirm their importance for IL-10 expression [114]. More surprisingly, this IL-10 competency may also be related to a marked decrease in Akt expression, with altered BCR signaling becoming constitutively active. Pathologically prolonged TCL1A expression in mice is associated with enhanced IRF4, BLIMP-1 and IL-10 expression, according to a mechanism that may not necessarily require the Akt pathway [114].

Other studies have also recently investigated pathways implicated in IL-10 production in Eµ-TCL1A mice. Alhakeem et al. first detected that BCR signaling was implicated in IL-10 expression by using inhibitors of Src, Syk and Btk kinase and thereby reducing IL-10 mRNA levels. They postulated that IL-10 expression was regulated at a transcriptional level. Using Syk inhibitors, they noted a decrease in Akt phosphorylation and postulated that the PI3K-Akt pathway may play a role in IL-10 expression [150]. They further investigated this subject in another study using the same kinase inhibitors and noted BCR involvement in IL-10 expression with a decrease in ERK1/2 phosphorylation. Investigating a potential downstream transcription factor that may be activated by ERK1/2, they found that Sp1 was the only transcription factor enhanced by BCR signaling and reduced by Syk inhibition. Moreover, they found that mithramycin A, a Sp1 inhibitor, reduced IL-10 protein levels in a dose-dependent manner. Additionally, their Chromatin Immunoprecipitation assays revealed an 8-fold enrichment in the binding of Sp1 to the IL-10 promoter in CD5^+^ B220^+^ B cells of Eµ-TCL1A mice. These results were further supported in primary human CLL samples in which Syk inhibitors reduced ERK1/2 phosphorylation and Sp1 and IL-10 levels [151]. These results were consistent with the implication of a particular BCR signaling pathway being activated in Eµ-TCL1A CD5^+^ B220^+^ B cells participating in IL-10 expression [114]. They were also consistent with other studies implicating Sp1 in IL-10 expression in regulatory T cells in response to GSK3 inhibition [138], also favoring IL-10 expression in B cells through BLIMP-1 [139]. Notably, Sp1 may regulate the expression of both IL-10 and TCL1A in B cells [81,151].

In addition to Akt coactivation, TCL1A acts as a transcriptional regulator. Pekarsky et al. showed that TCL1A physically interacts with several factors of the transcriptional complex AP-1, notably c-Jun and c-Fos, and acts as an inhibitor of the AP-1 complex [99]. TCL1A also represses the expression of truncated receptor-type protein tyrosine phosphatase (PTPROt), a phosphatase expressed in naïve B cells that represses lymphoma in Eµ-TCL1A mice, and regulates BCR signals by modulating the activity of tyrosine kinases, including Lyn, Syk and ZAP70 [152,153,154,155]. The inhibition of PTPROt occurs early and at the transcriptional level. PTPROt transcription is dependent on the AP-1 complex elements c-fos and c-jun, and TCL1A impairs AP-1 complex activity by repressing c-fos and inhibiting c-jun phosphorylation. These results were confirmed in B-CLL cell lines and primary cells, showing that *PTPROt* promoter activity depends on the AP-1 complex and that its inhibition is correlated with TCL1A expression [155]. Using a double transgenic mouse model overexpressing PTPROt and TCL1A, the same team that discovered TCL1A repression of PTPROt highlighted that PTPROt overexpression had a tumor suppressor effect in vivo, which had been previously identified in other studies [153,156]. In contrast, this same group studied the effect of PTPROt downregulation in Eµ-TCL1A mice by inactivating a single (heterozygous knockout [HET]) or both (homozygous knockout [ROKO]) PTPROt alleles in wild-type (WT) and Eµ-TCL1A mice to determine how it impacts CLL onset in these mice. Deleted PTPROt did not alter BCR signaling pathways, as indicated by WT, HET and ROKO mice presenting similar basal levels of Lyn, Syk, p-Akt and p-ERK. In Eµ-TCL1A mice, PTPROt may be considered a tumor repressor when one allele is deleted and, unexpectedly, may be a tumor promoter when both alleles are deleted [157]. IL-10 expression was higher in the tumor cells from the HET/Eµ-TCL1A mice and even higher in the ROKO/Eµ-TCL1A mice than in WT/Eµ-TCL1A mice, suggesting that PTPROt decreases enhanced IL-10 expression. In tumor cells, higher expression of IL-10R1 was observed in HET and ROKO/Eµ-TCL1A mice, consistent with higher phosphorylation of STAT3 activated by the IL-10 receptor [157]. In addition, this finding needs to be specifically extended to understand how PTPROt expression may influence that of IL-10; it may also have a link to the higher IL-10 potency found in WT/TCL1A mice [113], which may be explained partially by TCL1A repression of PTPROt and acquisition of a B10 phenotype.

## 7. Conclusions

In summary, we first see that TCL1A may participate in the increase in the proportion of immature B cells in TOL patients, considering its effects on B cell survival, proliferation, growth and differentiation. Several studies also pointed to a potential link between TCL1A expression and the development of a Breg phenotype, notably through IL-10 expression. However, TCL1A expression in mice is not mandatory for the development of B cell IL-10 competency since some Eµ-TCL1A mice retained normal frequencies and absolute numbers IL-10 competent B cells [113], with a B cell clonal variation in their basal levels of the IL-10 produced [150]. Moreover, mature subsets of IL-10 competent B cells, such as CD24^hi^ CD27^+^ memory [158] and CD19^+^ CD24^hi^ CD27^int^ plasmablasts in humans [135], are known to physiologically repress TCL1A. This also indicates that TCL1A expression is dispensable for IL-10 B cell competency. Nonetheless, TCL1A overexpression in human and mouse B cells strongly correlates with an enhancement of IL-10^+^ B cells. TCL1A may favor orientation of B cells to an IL-10 production phenotype, by a mechanistic remaining elusive, but needed to be considered notably in immature B cell populations of tolerant renal transplant patients expressing both TCL1A and IL-10. In addition to these observations, TCL1A overexpression in B cells from TOL patients may also be derived from an independent increase in immature B cells, which are known to express this protein. More investigation is thus required to determine whether TCL1A overexpression represents a condition suitable for the development of renal transplantation tolerance. This will be deciphered using in vitro models of induction of regulatory B cells, such as B10 and GZMB-producing cells, from TOL and no-TOL kidney transplant patients. In these cells, TCL1A interactions described in this work would systematically be analyzed at transcriptomic and proteomic levels regarding to their effects towards regulatory process, including IL-10 expression. The use of animal models described above in the settings of transplantation models will also evidence TCL1A’s key role in immune mechanisms. These experiments would provide a clearer picture of TCL1A involvement in kidney tolerance and foster its use as biomarker or potential therapeutic target.

## Figures and Tables

**Figure 1 cells-10-01367-f001:**
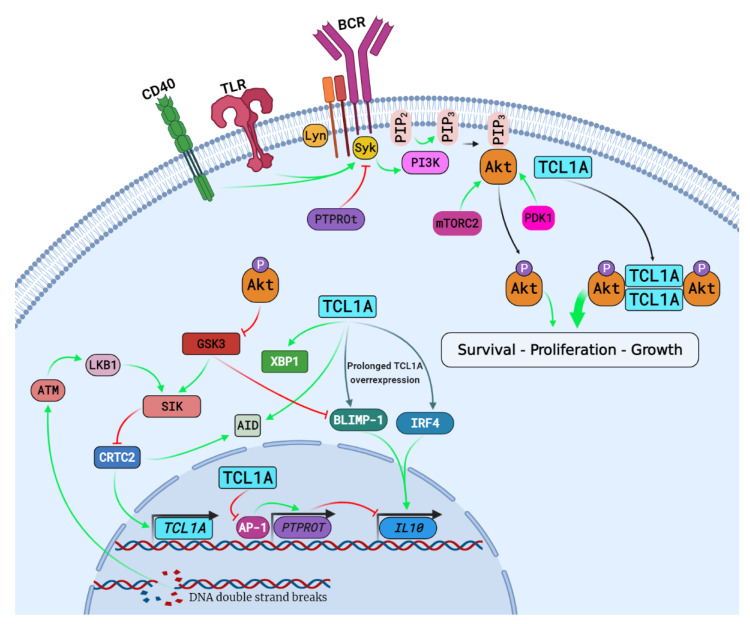
Schematic representation of TCL1A interactions in B cells. Stimulation of surface B cells’ receptors (CD40, TLR, BCR) induces a phosphorylation cascade implying, notably, Lyn and Syk enzymes leading to PI3K activation. PI3K converts PIP2 into PIP3, allowing Akt recruitment to the membrane and its full activation by mTORC2 and PDK1. Once phosphorylated, Akt activates downstream targets leading to B cell survival, growth, and proliferation. TCL1A homodimers complex with Akt allowing to increase its effects by favoring its phosphorylated state and its translocation to the nucleus. During GC transformations, AID generates DNA double stand breaks inducing CRTC2 inactivation by SIK through ATM and LKB1 leading to AID and TCL1A downregulation. TCL1A overexpression could lead to XBP1 and AID upregulation and in the long term of BLIMP-1 and IRF4, both transcription factors involved in B cell differentiation and IL-10 expression. TCL1A also impairs PTPROt expression through AP-1 inhibition while PTPROt downregulation increase IL-10 expression. Phosphorylated Akt also inactivate GSK3 which favors IL-10 expression through BLIMP-1. Created with BioRender.com (accessed on 20 May 2021).

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
