# Peer review of "TCL1A, B Cell Regulation and Tolerance in Renal Transplantation"

_cells, 2021, doi:10.3390/cells10061367_

Round 1
Reviewer 1 Report
This is a mammoth and impressive work reviewing B cell signalling mechanisms, phenotype and function, not limited to the context of transplantation.
There are a number of passages that require grammatical attention. This mostly pertains to the Introductory and Concluding Sections of the paper. Here, the sentences are quite long-winded and alterations may assist with clarity of ideas: eg lines 65-70, line 73, lines 536-540.
In the section under "a ....biomarker of tolerance process", clarity would also be assisted by referring to the organ involved in each study, which would avoid the reader to have to look up each study to determine if relevant to their organ of interest. This information has been captured in some parts (eg line 183) but mostly not in other paragraphs.
As a transplant clinician I would also like to see a comparison of TCL1A versus other B cell markers in discovery and/or validated gene sets as an example of the relative importance of this gene over others in the B cell repertoire.
Author Response
Dear Reviewer,
Thank you very much for your reviewing work.
Question 1
There are a number of passages that require grammatical attention. This mostly pertains to the Introductory and Concluding Sections of the paper. Here, the sentences are quite long-winded and alterations may assist with clarity of ideas: eg lines 65-70, line 73, lines 536-540.
According to your comments about long sentences, we modified sentences at manuscript locations that you pointed, highlighted in yellow.
Question 2
In the section under "a ....biomarker of tolerance process", clarity would also be assisted by referring to the organ involved in each study, which would avoid the reader to have to look up each study to determine if relevant to their organ of interest. This information has been captured in some parts (eg line 183) but mostly not in other paragraphs.
Concerning paragraph “TCL1A, a biomarker of tolerance process”, we mentioned the transplanted organ in each study according to your comment.
Question 3
As a transplant clinician I would also like to see a comparison of TCL1A versus other B cell markers in discovery and/or validated gene sets as an example of the relative importance of this gene over others in the B cell repertoire.
Finally, concerning TCL1A expression level in relation to other B cell related biomarkers, studies evoked in this work has been looking at the significance of difference of expression of the genes between the different groups of patients (TOL, HVs, CR). But since relative gene expression was used (in contrast to absolute quantification), it is not possible to postulate on expression level differences between different genes for each patient. As an example it is not possible to say if the gene TCL1A is more expressed than the gene CD19. Nevertheless, according to B-cell related markers, we emphasized that TCL1A was repeatedly found in B-cell related signatures, along with MS4A1 (coding for CD20), CD19, CD79B, CD40 and/or BANK1 depending on studies. We have added information along these lines in the review.
Please see the attachment
Modifications are underlined in yellow in the manuscript.
We hope these corrections would please you.
Kind regards

Reviewer 2 Report
This paper discusses TCLA1 in B cells which have potential roles in tolerance. Overall, the information represents valuable information regarding biomarkers for renal transplantation tolerance. I recommend that this paper be accepted after minor revision.
Minor comments.
- The figure 1 is blurry. Please replace it with a clear picture.
- Please suggest a more specific future study design to elucidate the role of TCL1A in renal transplantation tolerance.
Author Response
Dear Reviewer,
Thank you very much for your reviewing work.
Question
1. The figure 1 is blurry. Please replace it with a clear picture.
According to your comment about the blurry figure, we used an image with a better resolution.
Question
2. Please suggest a more specific future study design to elucidate the role of TCL1A in renal transplantation tolerance.
According to your second comment, we indicated in the conclusion part (p. 12; l. 517-522) main experiments that we are currently conducting and that would allow to investigate on TCL1A role in B cell biology and regulatory mechanisms according to interactions pointed in this work: “This will be deciphered using in vitro models of induction of regulatory B cells, such as B10 and GZMB-producing cells, from TOL and no-TOL kidney transplant patients. In these cells, TCL1A interactions describes in this work would systematically be analyzed at a transcriptomic and proteomic levels regarding to their effects towards regulatory process, including IL-10 expression. The use of animal models described above in the settings of transplantation models will also evidence TCL1A key role in immune mechanisms. These experiments would provide a clearer picture of TCL1A involvement in kidney tolerance and foster it’s used as biomarker or potential therapeutic target.”
Please see the attachement
Modifications are underlined in yellow in the manuscript.
We hope these corrections would please you.
Kind regards
